# Corticosterone Administration Alters White Matter Tract Structure and Reduces Gliosis in the Sub-Acute Phase of Experimental Stroke

**DOI:** 10.3390/ijms22136693

**Published:** 2021-06-22

**Authors:** Katarzyna Zalewska, Rebecca J. Hood, Giovanni Pietrogrande, Sonia Sanchez-Bezanilla, Lin Kooi Ong, Sarah J. Johnson, Kaylene M. Young, Michael Nilsson, Frederick R. Walker

**Affiliations:** 1School of Biomedical Sciences and Pharmacy, University of Newcastle, Callaghan 2308, Australia; kasia.zalewska@health.qld.gov.au (K.Z.); rebecca.hood@newcastle.edu.au (R.J.H.); g.pietrogrande@uq.edu.au (G.P.); Sonia.SanchezBezanilla@uon.edu.au (S.S.-B.); 2Priority Research Centre for Stroke and Brain Injury, University of Newcastle, Callaghan 2308, Australia; sarah.johnson@newcastle.edu.au (S.J.J.); michael.nilsson@newcastle.edu.au (M.N.); 3Hunter Medical Research Institute, Newcastle 2305, Australia; 4NHMRC Centre of Research Excellence Stroke Rehabilitation and Brain Recovery, Newcastle 2305, Australia; 5School of Pharmacy, Monash University Malaysia, Bandar Sunway 47500, Malaysia; 6School of Electrical Engineering and Computing, University of Newcastle, Callaghan 2308, Australia; 7Menzies Institute for Medical Research, University of Tasmania, Hobart 7000, Australia; Kaylene.Young@utas.edu.au

**Keywords:** white matter tracts, stroke recovery, oligodendrocyte, myelin, stress, corticosterone, glia

## Abstract

White matter tract (WMT) degeneration has been reported to occur following a stroke, and it is associated with post-stroke functional disturbances. White matter pathology has been suggested to be an independent predictor of post-stroke recovery. However, the factors that influence WMT remodeling are poorly understood. Cortisol is a steroid hormone released in response to prolonged stress, and elevated levels of cortisol have been reported to interfere with brain recovery. The objective of this study was to investigate the influence of corticosterone (CORT; the rodent equivalent of cortisol) on WMT structure post-stroke. Photothrombotic stroke (or sham surgery) was induced in 8-week-old male C57BL/6 mice. At 72 h, mice were exposed to standard drinking water ± CORT (100 µg/mL). After two weeks of CORT administration, mice were euthanised and brain tissue collected for histological and biochemical analysis of WMT (particularly the corpus callosum and corticospinal tract). CORT administration was associated with increased tissue loss within the ipsilateral hemisphere, and modest and inconsistent WMT reorganization. Further, a structural and molecular analysis of the WMT components suggested that CORT exerted effects over axons and glial cells. Our findings highlight that CORT at stress-like levels can moderately influence the reorganization and microstructure of WMT post-stroke.

## 1. Introduction

White matter tract (WMT) integrity post-stroke is an area of significant interest, as it is a parameter that correlates with long-term recovery potential [1]. WMTs can be directly damaged at the time of stroke, but are also affected by secondary Wallerian degeneration [2,3]. Wallerian degeneration is characterised by the progressive degeneration of axons, followed by the infiltration of resident macrophages and infiltrating monocytes, which ultimately results in the degradation of myelin and atrophy of the WMTs [4]. The association between stroke and secondary Wallerian degeneration of white matter has been explored in a number of preclinical [2,5,6,7,8] and clinical studies [9,10]. Neuroimaging studies suggest that these changes are associated with neurological deficits and poorer functional outcomes post-stroke [11,12,13]. A study by Jokinen et al. (2005) found that WMT hyperintensities on T2 diffusion-weighted images could be used as independent predictors of post-stroke cognitive impairment and were associated with poor performance in tests of mental speed, executive function, memory, and visuospatial functions [13]. Investigations into WMT hyperintensities and cognitive deficits suggest that the location, severity, and number of hyperintensities may all be important factors that affect functional outcome [14,15].

Stress is a major and persistent aspect of stroke recovery [16,17,18]. A number of clinical studies have identified that greater levels of perceived stress in the acute phase post-stroke (14 days post-stroke) were associated with worse outcomes [19,20]. High levels of post-stroke stress are associated with elevated levels of cortisol, a stress biomarker that is associated with increased dependency, morbidity, and mortality [19,21,22]. There is also extensive pre-clinical evidence showing that chronic stress after stroke enhances neuron loss, accumulation of neurotoxic proteins, and inhibits brain repair processes (including those at sites of secondary neurodegeneration), leading to worse outcomes [23,24,25]. Our group has extended these findings to show that exogenous corticosterone (CORT; the rodent equivalent of cortisol) administration post-stroke significantly affects the glial response to stroke when administered at stress-like levels within the first 14 days post-stroke [26,27]. As the structural integrity of WMTs is sensitive to chronic stress [28,29], it is possible that WMT changes occurring post-stroke are modified by cortisol/ CORT. Evidence in support of this comes from a study by Magalhães et al. (2017), who demonstrated that 15 days of inescapable (chronic) stress significantly disrupted the structure of several WMT bundles. This maladaptive stress response was associated with increased CORT levels [28]. Clinical studies seem to confirm the link between cortisol and WMT structure in neurodevelopment and neurological conditions [30,31,32]. However, to date, the impact of cortisol/CORT on WMT changes after stroke has not been studied.

Herein, we examine changes that occur within the corpus callosum (CC) and corticospinal tract (CST) of mice following a photothrombotic stroke, and investigate the influence that high levels of CORT on white matter pathology. We quantify tissue loss and structural changes in these WMTs and evaluate changes in key oligodendrocyte proteins, as well as axon integrity and degeneration. We additionally examine the extent of gliosis in the same regions, as gliosis has been shown to be associated with remodelling of WMT after a stroke [33]. We hypothesised, that CORT administration after a stroke would significantly exacerbate white matter pathology. Specifically, we investigated tissue loss, changes in myelin structure (Cresyl violet, myelin basic protein; MBP, and Sudan Black staining), and axonal degeneration (Fluoro-Jade C labelling; FJC). We also assessed changes in markers of astrogliosis (glial fibrillary acidic protein; GFAP) and microgliosis (cluster of differentiation 11b; CD11b, and cluster of differentiation 68; CD68), as well as markers of mature oligodendrocytes and myelin (MBP and Claudin-11) and oligodendrocyte precursor cells (neuroglial antigen 2; NG2).

## 2. Results

### 2.1. Oral CORT Administration Post-Stroke Exacerbates Tissue Loss

Cresyl violet staining was performed to examine the effect of CORT on tissue loss. Animals exposed to stroke (stroke and stroke + CORT) had significantly more tissue loss from the somatosensory and motor cortices compared to sham animals (sham, sham + CORT) (*p* < 0.05; Figure 1). However, at Bregma −2.18, the tissue loss in the stroke-only animals was not different to either sham group. There was no effect of CORT administration on tissue loss in the sham animals. Following stroke, however, treatment was found to exacerbate tissue loss at Bregma levels 0.02 (stroke = 14.75%, stroke + CORT = 21.04%; *p* < 0.05) and −0.94 (stroke = 11.87%, stroke + CORT = 18.83%; *p* < 0.05). In the stroke + CORT group, direct damage caused by the stroke induction to the CC was observed.

### 2.2. Oral CORT Administration Alters the Structure of the Corpus Callosum (CC) and Corticospinal Tract (CST)

We next examined white matter loss from two major WMTs, the CC and the CST using Cresyl violet staining. Callosal area (mm^2^) was measured at Bregma levels 0.62, 0.02 and −0.94, within defined regions on the contralateral and ipsilateral hemispheres (Figure 2C,D). These data were used to calculate contralateral callosal area, relative to that of the ipsilateral hemisphere (Figure 2A,B). In sham and sham + CORT mice, the mean fold change (FC) in area was close to 1, indicating that the callosal area was consistent between hemispheres at all three Bregma levels. In contrast, the contralateral callosal area was significantly smaller than the ipsilateral area in stroke-only mice. Callosal area was directly affected by the stroke induction. When compared with sham controls, the ratio between callosal areas was significantly different in the stroke-only group at all investigated Bregma levels (Figure 2A; level 0.62 FC = 0.7089, *p* < 0.001; level 0.02 FC = 0.776, *p* < 0.01; level −0.94 FC = 0.5716, *p* < 0.001). Surprisingly, the effect of CORT administration after stroke was highly variable in the callosal area at the different Bregma levels (Figure 2A). Following stroke, CORT administration had no effect on the stroke-mediated decrease in contralateral callosal area at Bregma −0.94, but it reduced the disparity between contralateral and ipsilateral callosal area at Bregma level 0.62 (Stroke group FC = 0.7089 vs. stroke + CORT FC = 0.8872, *p* < 0.01) and exacerbated the difference at Bregma 0.02 (for Stroke group FC = 0.776 vs. for Stroke + CORT FC = 0.6116, *p* < 0.05).

In order to investigate remote changes in the WMT structure, we analysed the CST, an area distal from the stroke injury. As per our analysis of the CC, a similar approach was used to assess changes in the area of the CST at Bregma levels −0.94, −1.58, and −2.18. We observed differences in the area of the CST only at −1.58 Bregma in the stroke-alone group compared to all other groups (Stroke FC = 1.329 vs. sham group FC = 1.044, *p* < 0.01; sham + CORT FC = 1.009 *p* < 0.001; stroke + CORT FC = 0.885, *p* < 0.001, Figure 2B).

### 2.3. Both CORT and Stroke Alter Myelin Structure

Qualitative analysis of the myelin structure labelled by MBP and Sudan Black staining indicated no observable differences in the CC and the dorsal part of the CST between any of the groups (Figure 2C and Appendix A). Of note, MBP staining showed clear axonal bundles were clearly notable in the ventral CST of sham animals but not observable in the other three treatment groups. Both stroke and CORT disrupted MBP-positive structures within CST (Figure 2D). In the sham group, Sudan Black analyses revealed clearly distinguished axonal structures in the ventral CST region, which were disrupted in the stroke and stroke + CORT groups. In the sham + CORT group, structures were still visible; however, axonal bundles were less profound.

### 2.4. Both CORT and Stroke Have no Effect on Total Level of MBP Protein

MBP in the CC: There was no difference in the protein level of MBP in the CC region between the four groups (Figure 3A).

MBP in the CST: A modest increase in MBP was observed in the stroke + CORT group (stroke FC = 0.8641 vs. stroke + CORT FC = 1.179, *p* < 0.05, Figure 3B). No significant changes were observed in the other groups.

### 2.5. Oral CORT Administration Reduces Expression of Claudin-11

Claudin-11 in the CC: A significant increase was observed after stroke and this effect was diminished by CORT administration (stroke FC = 3.078 vs. stroke + CORT FC = 0.6784, *p* < 0.001, Figure 3A). There were no significant differences between sham and sham + CORT.

Claudin-11 in the CST: Stroke did not affect the expression of Claudin-11 in the CST. However, both CORT groups (sham + CORT and stroke + CORT) displayed significantly reduced levels of Claudin-11 (sham FC = 1 vs. sham + CORT FC = 0.2947, *p* < 0.05 and stroke FC = 0.8917 vs. stroke + CORT FC = 0.2596, *p* < 0.05, Figure 3B).

### 2.6. Stroke Induction Reduces Level of NG2

NG2 in the CC: Stroke induced a significant decrease in the expression of NG2 in the CC region (sham FC = 1 vs. stroke FC = 0.5803, *p* < 0.01, Figure 3A). There was no significant effect of CORT administration in either recipient group when compared with the respective stroke and sham control groups in the post hoc analyses.

NG2 in the CST: There was no significant effect of CORT or stroke in CST region.

### 2.7. Oral CORT Administration Reduces the Expression of Microglial and Astrocytic Markers

Changes in the expression of microglial and astrocytic markers were analysed using cumulative thresholding, as per [34]. Analyses were performed on two regions within CC and two regions within CST (Figure 4A,C, left panels and Appendix A).

CD68 expression in the CC: We identified that a significant effect in the medial CC (F_Stroke(1, 25)_ = 23.39, *p* < 0.001), with an increase in the expression level compared to sham group (sham = 2.305% vs. stroke = 64.3%, *p* < 0.001). In the same region, we also noted a significant effect for CORT (F_CORT(1, 25)_ = 4.857, *p* < 0.05). CORT administration after stroke was observed to reduce the expression of CD68 approximately threefold compared to the stroke-alone condition (stroke = 64.3% vs. stroke + CORT = 25.84%, *p* < 0.05). In the lateral CC, which was more distant from the primary occlusion site, we again observed a stroke-induced increase in CD68 expression, an effect that was significantly lower in the stroke + CORT group (sham = 5.713% vs. stroke = 21.24%, *p* < 0.05, stroke vs. stroke + CORT = 4.961%, *p* < 0.05).

CD68 expression in the CST: In the dorsal-most region of the CST, stroke significantly increased the intensity of CD68 expression, and this effect was significantly reduced in the stroke + CORT group (sham = 2.703% vs. stroke = 21.96%, *p* < 0.01; stroke vs. stroke + CORT = 5.223%, *p* < 0.01). Less profound changes were observed in the ventral region of the CST (sham = 2.343% vs. stroke = 12.21%, *p* < 0.001; stroke vs. stroke + CORT = 2.147%, *p* < 0.001).

CD11b expression in the CC: Analysis of CD11b from the CC was undertaken using Western blot analysis. Stroke increased the expression of CD11b, and this enhancement was significantly lower in the stroke + CORT animals (sham FC = 1 vs. stroke FC = 1.572, *p* < 0.05 and stroke vs. stroke + CORT FC= 1.004, *p* < 0.05, Figure 4B).

CD11b expression in CST: A significant increase in protein level of CD11b was noted in the CST in the stroke group relative to the sham group, but no effect was observed for CORT-treated animals (Figure 4D).

GFAP expression in the CC: Both CC regions (medial and lateral) exhibited stroke-induced increases in GFAP expression. Stroke + CORT animals had significantly lower levels when compared to the stroke-alone group (medial region—proximal to the stroke induction; sham = 30.75% vs. stroke= 80.92%, *p* < 0.001; stroke vs. stroke + CORT= 27.2%, *p* < 0.001); (lateral region—distal from stroke induction; sham = 10.49% vs. stroke= 30.86%, *p* < 0.01; stroke vs. stroke + CORT = 1.254%, *p* < 0.001, Figure 4A). Western blot results were consistent with immunohistochemical analysis, with stroke elevating GFAP levels and CORT administration suppressing them (sham FC = 1 vs. stroke FC = 2.5, *p* < 0.001; stroke vs. stroke CORT FC = 1.004, *p* < 0.05, Figure 4B).

GFAP expression in the CST: Both CST regions (dorsal and ventral) exhibited stroke-induced increases in GFAP, an effect that was significantly reduced in stroke + CORT animals treated with CORT (dorsal CST; sham = 11.91% vs. stroke = 28.19%, *p* < 0.01; stroke vs. stroke + CORT = 3.367%, *p* < 0.001; ventral CST; sham = 17.54% vs. stroke = 44.24%, *p* < 0.01; stroke vs. stroke + CORT = 15.92%, *p* < 0.01). Western blot analysis of changes in GFAP within the CST indicated that stroke increased GFAP expression in the CST (sham FC = 1 vs. stroke FC = 1.305, *p* < 0.05). However, neither of the CORT recipient groups showed an effect on GFAP expression (Figure 4D).

### 2.8. Axonal Degeneration is Enhanced in Stroke Animals Treated with CORT in the CST; However, Not the CC

FJC staining was used to visualise axonal degeneration and was performed at Bregma level ~0.02 for CC and at −1.58 for CST. Observational analyses of FJC in the CC clearly revealed degeneration of fibre tracts in both stroke groups (stroke and stroke + CORT). There was no observable difference in the CC between sham and sham + CORT (Figure 5B). Scattered FJC-positive fibre tracts were visible within the CST in the sham + CORT, stroke, and stroke + CORT groups (Figure 5D).

## 3. Discussion

In the current study, we examined the effect of orally delivered CORT on white matter using a number of approaches. The primary aim of this study was to investigate the effects of CORT when delivered at stress-like concentrations on WMT pathology after experimental stroke. Specifically, we were interested in examining changes in two major regions, the CC and the anterior CST, in order to investigate both direct and distant effects of stroke ± CORT. Using a combination of approaches, including immunohistochemistry and Western blotting, we were able to make a number of important observations. Our findings regarding changes in WMTs indicated that, while there were some changes evident in the white matter, the effects were quite modest, marker specific, and regionally specific.

First, we showed that orally administered CORT exacerbated gross tissue loss in the ipsilateral hemisphere post-stroke. In addition to gross changes, we specifically considered changes in the volume of the CC and CST. This analysis indicated a complex series of changes which were dependent upon the rostro–caudal level considered. For the CC, we observed a clear effect of stroke on WMT volume across the investigated levels. Interestingly, we identified that post-stroke CORT treatment reduced the amount of tissue loss at 0.62 mm but increased it at 0.02 mm Bregma compared with the stroke-only group. In the CST, stroke did not result in tissue loss at any level; in fact, at −1.58 mm Bregma, the area was significantly higher than all the other groups. Interestingly, CORT prevented this increase but, importantly, the levels were no different to the sham levels. Together, these results indicate that the impact of CORT on white matter thinning is not consistent across the rostro–caudal axis. Moreover, the size of the effect was relatively small in comparison to the effect that stroke had on thinning, particularly in relation to changes within the CC.

Next, we considered the effect of CORT on the individual components of white matter: axons and glial cells. First, we investigated changes in the distribution of labelling for MBP and Sudan Black within the WMT. MBP is considered to be the main marker of axonal myelinization, which identifies myelin in both the central and peripheral nervous systems and also stains the cytoplasm of oligodendrocytes and Schwann cells. Our visual qualitative observation of MBP indicated no obvious changes in the distribution of MBP within the CC. We did note some modest evidence of changes in the amount of MBP bundling in the CST. These changes were also largely mirrored in the distribution of Sudan Black labelling, which is a non-ionic azo dye that accumulates mostly in lipids, which are abundant in myelin sheaths. Additionally, we evaluated the possibility of axonal degeneration. We used FJC, which is considered a marker of axonal degeneration and routinely used in acute and chronic neurodegeneration studies [35]. We observed robust FJC labelling in both stroke groups but no obvious differences were observable in the overall levels between the control groups and the CORT-treated animals. Interestingly, FJC staining revealed CORT delivery alone was sufficient to induce degeneration of axon fibres in the CST, but not in the CC. This intriguing finding suggests that just 2 weeks of elevated CORT levels is sufficient to induce axonal degeneration in WMTs. Further, we considered changes in the expression of MBP, the tight junction protein Claudin-11, and NG2, a marker for oligodendrocyte precursor cells, using Western blotting. Here, our analysis revealed that CORT had no effect on NG2 levels, no effect on the level of MBP protein in the CC, but enhanced levels with the CST and decreased Claudin-11 expression in both the CC and CST. The lack of changes in MBP expression level observed in this study was consistent with other neurodegenerative paradigms. For instance, in experimental autoimmune encephalomyelitis, the total level of MBP is not changed but is instead restructured and aggregated [36]. We found that the changes in Claudin-11 do not completely mirror the results of NG2. The findings suggest that mature oligodendrocytes are more responsive to hypothalamic–pituitary–adrenal (HPA) axis dysregulation than their precursor cells [37]. This finding of CORT affecting oligodendrocytes and WMT is important, as both CORT and stress have been shown to promote oligodendrogenesis in the dentate gyrus [38]. Stress has also been shown to decrease retrieval after extinction and alter neuronal activity in the prelimbic and infralimbic cortices [39]. Our results suggest that, in the CST, CORT promotes the opposite action than in the dentate gyrus. This further supports our finding that the effect of glucocorticoids is region specific.

Lastly, we assessed the effect of CORT on micro- and astroglial markers. Specifically, we identified that CORT reduced the expression of CD68 and GFAP within the CC, an effect that was supported with our Western blot analysis. The effect was also evident within the CST by immunohistochemistry but not supported by Western blot. Some discrepancies between immunohistochemistry and Western blot are not surprising, as Western blot allows for the investigation of global protein changes in the region which can sometimes be associated with lower sensitivity to specific regional changes. The observed increase in gliosis within the CST was almost 5x smaller than in CC, as indexed by immunhoistochemistry; therefore, it is likely that the CORT-induced decrease in gliosis was regionally specific.

Our research showed some effect of CORT administration on WMT after stroke; however, it is not clear whether this effect is strictly negative. Clinically, WMT reorganization and microstructure have been proven to be associated with cognitive decline and functional recovery after stroke [11,12,13,40,41]. Similar effects after stroke have been observed in patients experiencing chronic stress and elevated cortisol levels [16,21,42]. In the current study, we were motivated by existing clinical literature suggesting a strong impact of high cortisol levels on WMT microstructure and integrity [30,31,32,43,44]. However, to our best knowledge there are currently no studies specifically aimed at investigating how an elevated CORT level affects WMT after stroke. Another important question is whether CORT administration would have the same effects in female mice. It is well known that there are sex differences in stroke severity and outcomes [45]. Additionally, a number of groups have documented gender differences in response to HPA axis activation, i.e., the stress response system. In general, glucocorticoid levels are higher in females than in males after HPA axis stimulation, and estradiol exerts modulating effects on HPA axis functions, including HPA axis responsiveness and sensitivity to glucocorticoid negative feedback [46]. Therefore, the effects of CORT administration on recovery after stroke could significantly vary between the sexes, and should be considered in future studies. Our research indicates that CORT impacts the structure of myelin within the CST, as well as decreases stroke-induced gliosis. Further studies are needed to determine the extent of CORT-induced myelin changes in these regions by using electron microscopy, for instance. Moreover, additional clinical research investigating the correlation between functional recovery, WMT structure and integrity, and cortisol level would be beneficial in the context of potential future therapies aimed at stress prevention after stroke.

## 4. Materials and Methods

### 4.1. Ethical Statement

All experiments were approved by the University of Newcastle Animal Care and Ethics Committee (A-2013-340) and conducted in accordance with the New South Wales Animal Research Act (1985) and the Australian Code of Practice for the Care and Use of Animals for Scientific Purposes (NHMRC).

### 4.2. Animals

C57BL/6 male mice (7 weeks old) were obtained from the Animal Services Unit at the University of Newcastle. Mice were housed in a temperature- (21 ± 1 °C) and humidity-controlled environment with food and water available ad libitum. Lighting was set to a 12:12 h reverse light cycle. Mice were acclimatised for 7 days prior to the start of the experiment.

### 4.3. Experimental Design

A total of 60 mice were randomly allocated to one of four groups: (1) sham (n = 14), (2) sham + CORT (n = 14), (3) stroke (n = 16), and (4) stroke + CORT (n = 16) (see Appendix A) before the commencement of any procedures. Within each group, mice were further randomised to either immunohistochemistry (fixed tissue analysis) or Western blot (fresh tissue analysis). Brains were collected at 17 days post-stroke.

### 4.4. Experimental Stroke and Oral Delivery of Corticosterone (CORT)

Mice assigned to the stroke groups were exposed to photothrombotic stroke, performed as previously described [24,47]. Briefly, anaesthetised animals (isoflurane 2% in 100% O_2_) were given an intraperitoneal injection of 0.2 mL Rose Bengal dye (10 mg/mL in sterile saline; Sigma-Aldrich, St. Louis, MO, USA) or 0.2 mL of vehicle (0.9% NaCl, Pfizer, Sydney, NSW, Australia) for sham animals. At 8 min post-injection, the skull was exposed and illuminated for 15 min using a cold light source (diameter 4.5 mm) positioned 2.2 mm lateral and 0 mm posterior to Bregma, targeting the left somatosensory cortex.

Corticosterone hemisuccinate (4-PREGNEN-11β, 21-DIOL-3, 20-DIONE 21-HEMISUCCINATE) was obtained from Steraloids Inc. (Newport, RI, USA). The drinking water solution was prepared so that the final concentration was 100 µg/mL, as described previously [26,27]. Briefly, CORT was stirred in pH 11–12 water solution (using NaOH) until completely dissolved, and then pH was adjusted to pH 7–7.5 (using HCl). The solution was changed every 72 h over the 14-day protocol. For control animals, drinking water was prepared by increasing and reducing pH without the addition of CORT. CORT administration began 72 h after the stroke for 14 consecutive days [26,27].

### 4.5. Tissue Processing

On day 17 post-stroke, mice were deeply anaesthetised by an intraperitoneal injection of sodium pentobarbital (Lethabarb, Virbac, Milperra, NSW, Australia, 325 mg/mL) prior to transcardial perfusion with ice cold saline (0.9%), followed by paraformaldehyde (4% *w*/*v* in saline, pH 7.4) for immunohistochemical analysis or saline-only for Western blot analysis. Brains were then extracted as fresh and fixed tissue, respectively, and examined to confirm the presence of a stroke and to confirm that it was located clearly within the somatosensory territory. Animals that did not meet these criteria were excluded at this stage of the study (see Appendix A). Fixed brains were post-fixed for 4 h in a 12.5% sucrose in 4% paraformaldehyde solution and then transferred to a 12.5% sucrose solution in 0.1 M PBS for cryoprotection and storage. Brains were sliced coronally on a freezing microtome at 30 μm (−25 °C, Leica, Waverly, Vic, Australia). Fresh brains were immediately frozen in −80 °C isopentane. Coronal sections were sliced at 200 μm on a cryostat (−20 °C, Leica, Waverly, Vic, Australia). Tissue was punched using a 2 mm tissue punch in the CC and CST regions. Samples were stored frozen at −80 °C until analysis.

### 4.6. Histology and Immunohistochemistry

Cresyl violet staining [48] was performed to confirm stroke and to examine the effect of CORT on tissue loss. Tissue loss was examined at 5 Bregma levels 0.62, 0.02, −0.94, −1.58 and −2.13 (Figure 1). To calculate tissue loss, we used the equation: [(area of contralateral hemisphere-area of ipsilateral hemisphere)/area of contralateral hemisphere] × 100% [26]. We next examined tissue loss from two major WMTs, the CC and the CST, using Cresyl violet staining. Cresyl violet is routinely used to identify the myelin in neural tissue [49]. Callosal area (mm^2^) was measured at Bregma levels 0.62, 0.02, and −0.94 within defined regions on the contralateral and ipsilateral hemispheres (Figure 2C,D). These data were used to calculate contralateral callosal area, relative to that of the ipsilateral hemisphere (Figure 2A,B).

Sudan Black B (Sigma-Aldrich, St. Louis, MO, USA) protocol was obtained from [50]. Briefly, sections were mounted and rinsed with 70% EtOH followed by 15 min incubation with Sudan Black B solution (600 mg of Sudan Black to 200 mL of 70% EtOH). After staining, sections were rinsed with 70% EtOH and water and were counterstained for 5 min with nuclear fast red solution (Sigma-Aldrich, St. Louis, MO, USA).

A FJC ready-to-dilute staining kit for identifying degenerating neurons (Biosensis, Thebarton, SA, Australia) was used as directed by the manufacturer. FJC staining was used to visualise axonal degeneration and was performed at Bregma level ~0.02 for CC and at −1.58 for CST.

For immuno-peroxidase labelling, free-floating tissue sections were immunostained as previously described [26,27]. Briefly, brain sections were incubated with 1% hydrogen peroxidase for 30 min at 25 °C followed by 3% horse serum for 30 min at 25 °C. Brain sections were incubated with specific primary antibodies: GFAP (mouse anti-GFAP, G3893; Sigma-Aldrich, St. Louis, MO, USA), CD68 (monoclonal rabbit anti-CD68, #76308; Abcam, Melbourne, Vic, Australia), and MBP (rabbit anti-MBP, #78896; Cell Signalling, Danvers, MA, USA) for 72 h at 4 °C, followed by secondary antibodies for 1 h at 25 °C and 2 h incubation at 25 °C with avidin–biotin-peroxidase complex, and finally developed using DAB peroxidase substrate. Brain sections were washed with PBS in between each incubation step.

### 4.7. Image Acquisition and Analysis

The following analyses were performed by a researcher blinded to the experimental conditions during the experiment. Histological and immunohistochemical analysis were performed as previously described [26,27,51]. Changes in the expression of astro- and microglial markers were analysed using cumulative thresholding, as per [34]. Briefly, brain regions were identified by reference to the mouse brain in stereotaxic coordinates. The mosaic images were cropped in CC and CST territories. For analysis of the acquired cropped images, we determined the pixel intensity that clearly reflected full inclusion of the immunolabelled signal. The individual pixel intensities at which this occurred were then determined and passed through to quantitative analysis.

### 4.8. Protein Extraction and Western Blotting

Protein was extracted from fresh tissue and Western blotting was performed, as previously described [24,26,27,51]. Briefly, protein samples were diluted with sample buffer (2% sodium dodecyl sulfate, 50 mM Tris, 10% glycerol, 1% DTT, 0.1% bromophenol blue, pH 6.8), followed by electrophoresis of 7.5 µg of total tissue protein in to Bio-Rad Criterion TGC Stain-Free 4–20% gels (Bio-Rad, Hercules, Ca, USA). Proteins were transferred onto PVDF membranes in transfer buffer (25 mM Tris, 200 mM glycine, and 20% methanol pH 8.3). Membranes were incubated overnight with the following primary antibodies: CD11b (rabbit anti- CD11b, #ab75476; Abcam, Melbourne, Vic, Australia), GFAP (mouse anti-GFAP, #3670; Cell Signalling, Danvers, MA, USA), Claudin-11 (rabbit anti- Claudin-11, RB215337; Thermo Scientific, Sydney, NSW, Australia), MBP (rabbit anti-MBP, #78896; Cell Signalling, Danvers, MA, USA), and NG2 (mouse monoclonal to NG2, ab129051; Abcam, Melbourne, Vic, Australia). On day 2, the membranes were incubated with the appropriate secondary antibodies: goat-anti-rabbit HRP (#170-6515; Bio-Rad, Hercules, Ca, USA) and goat anti-mouse HRP (#170-6516, Bio-Rad, Hercules, Ca, USA). Membranes obtained during Western blotting procedures were visualised using Amersham Imager 600 (GE Healthcare, Buckhinghamshire, UK) with Luminata Classico/Forte Western blotting detection reagents (Merck, Bayswater, Vic, Australia). The density of the bands was measured using Amersham Imager 600 Analysis Software (GE Healthcare, Buckhinghamshire, UK) and compared against normalising control beta-actin (mouse anti-beta-actin-HRP, #A38543; Sigma-Aldrich, St. Louis, MO, USA).

### 4.9. Statistical Analyses

All data are expressed as mean ± SEM and were analysed using GraphPad Prism v. 7.02 (La Jolla, Ca, USA). An a priori sample size calculation was performed on preliminary data investigating the relationship between CORT and GFAP expression in stroke animals. With an effect size of 5.9 and a standard deviation of 2.81, the sample size required to ensure that a treatment effect was detected was 6 animals per group. Two-way ANOVA was used to determine whether there were time and treatment effects across the four groups. Post hoc Tukey’s multiple comparisons tests were performed to analyse differences between the means of the groups. The significant differences shown on the graphs with asterisks (*) refer to the post hoc tests. Statistical significance was accepted at *p* < 0.05.

## 5. Conclusions

We have shown that both CORT administration and cortical ischemia induce morphological alterations in WMT structure and concomitant axonal degeneration within the CST. Furthermore, CORT administration after a stroke decreased stroke-induced gliosis, potentially by influencing WMT reorganization. While results from this study do not definitively answer the question as to whether CORT inhibition benefits stroke survivors, it shows proof of concept that CORT can worsen WMT outcomes post-stroke. These findings provide very strong indications that we should further investigate cortisol levels and WMT structures via both clinical and pre-clinical research. The main focus of future research should be placed on myelin structure rather than the level of proteins, as well as how oligodendrocytes respond to stressful conditions.

## Figures and Tables

**Figure 1 ijms-22-06693-f001:**
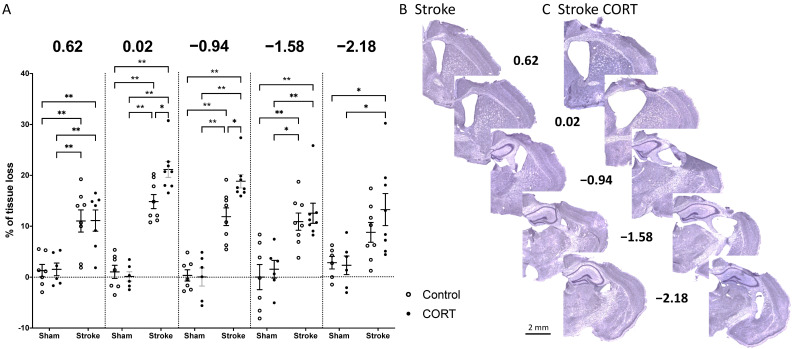
Corticosterone (CORT) increases tissue loss after stroke: (**A**) Percentage of tissue loss at five different Bregma levels across the four groups: sham, sham + CORT, stroke, stroke + CORT. The representative images show Cresyl violet staining of the same five Bregma levels in (**B**) representative stroke brain at 17 days post-stroke or (**C**) representative stroke + CORT brain at 17 days post-stroke with 14 days of CORT treatment. Data on the graph are expressed as mean ± SEM ** *p* < 0.01, * *p* < 0.05.

**Figure 2 ijms-22-06693-f002:**
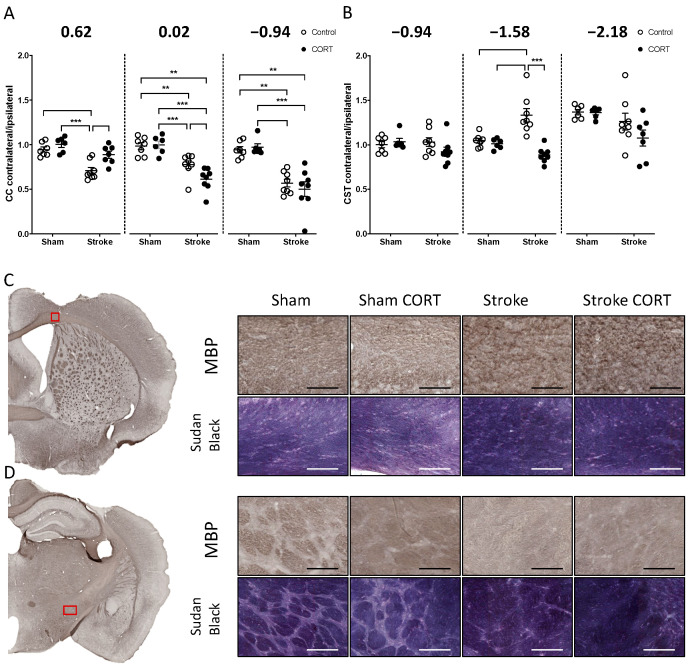
Changes in myelin structure after stroke and corticosterone (CORT) administration. Graphs showing the ratio of contralateral to ipsilateral volume of (**A**) the corpus callosum (CC) and (**B**) the corticospinal tract (CST) at 3 different Bregma levels. Representative immunohistochemical staining of myelin basic protein (MBP) in (**C**) the CC and (**D**) CST of the stroke affected hemisphere. The red squares represent the white matter regions used for analysis. The right panels show MBP and Sudan Black staining from the cropped regions across the four different groups. Scale bar in CC panels = 75 µm for MBP and 100 µm for Sudan Black. Scale bar in the CST panels = 100 µm for MBP and 150 µm for Sudan Black. Data on the graphs are expressed as mean ± SEM *** *p* < 0.001, ** *p* < 0.01.

**Figure 3 ijms-22-06693-f003:**
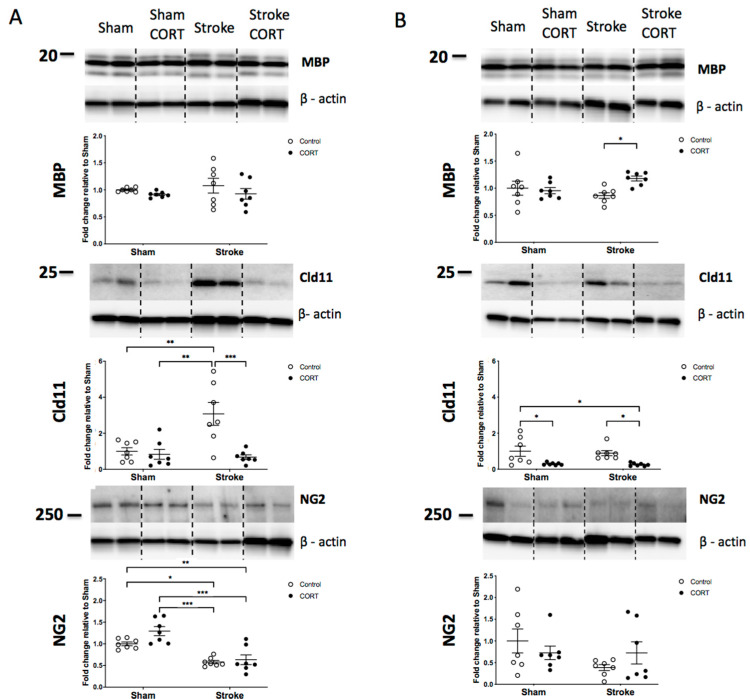
Protein expression after stroke and corticosterone (CORT) administration. Both panels show representative Western blots and quantitation of white matter tract specific proteins: myelin basic protein (MBP), Claudin-11 (Cld11) and neuroglial antigen 2 (NG2) in (**A**) the corpus callosum (CC) and (**B**) the corticospinal tract (CST). Data on graphs are expressed as mean ± SEM *** *p* < 0.001, ** *p* < 0.01, * *p* < 0.05.

**Figure 4 ijms-22-06693-f004:**
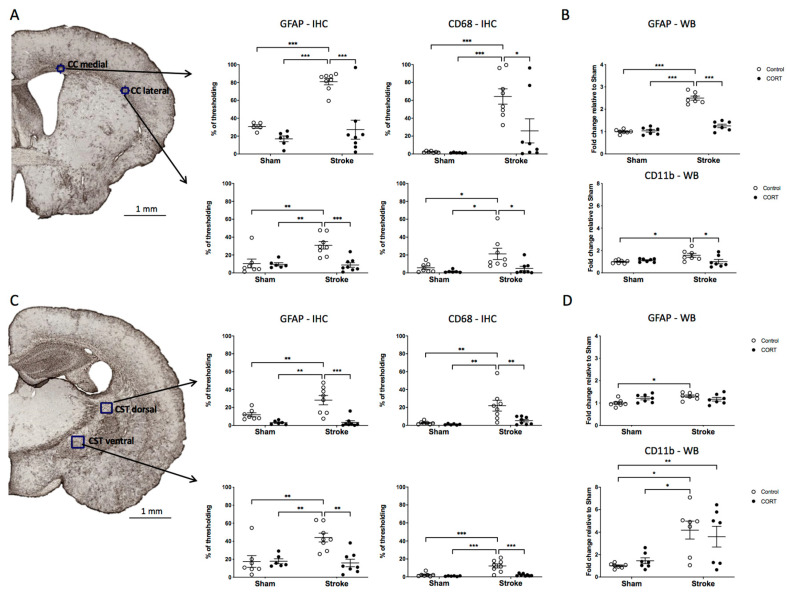
Astrogliosis and microgliosis in the corpus callosum (CC) and corticospinal tract (CST). The left side of Panel (**A**) shows representative glial fibrillary acidic protein (GFAP) immunohistochemistry of the CC. The graphs on the right side show changes in the percentage of thresholding for GFAP and cluster of differentiation 68 (CD68) for two regions of the CC (black boxes). (**B**) Western blot quantification of GFAP (upper graphs) and cluster of differentiation 11b (CD11b) (lower graph). The left side of panel (**C**) shows representative GFAP immunohistochemistry of the CST. Graphs on the right show changes in percentage of thresholding for GFAP and CD68 for 2 regions of CST (black boxes). (**D**) Western blot quantification of GFAP (upper panel) and CD11b (lower panel). Data on graphs are expressed as mean ± SEM *** *p* < 0.001, ** *p* < 0.01, * *p* < 0.05.

**Figure 5 ijms-22-06693-f005:**
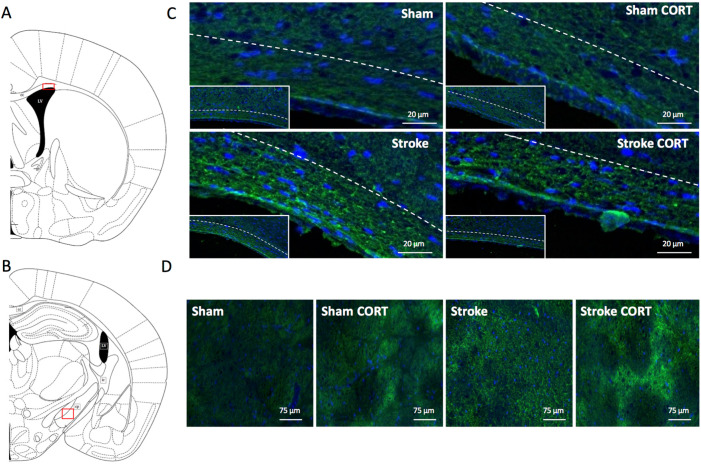
Corticosterone (CORT) and stroke induce axonal degeneration labelled by Fluoro-Jade C. The left panels show schematic representations of the approximate regions of the (**A**) corpus callosum (CC) and (**C**) corticospinal tract (CST) where pictures were taken. Panel (**B**) shows Fluoro-Jade C staining in the CC region across four groups: sham, sham + CORT, stroke, stroke + CORT. Small rectangles show the total area of CC and the big rectangles are a magnification of these areas. Scale Bar = 20 µm. Panel (**D**) shows representative pictures taken in the area of CST across all 4 groups. Scale Bar = 75 µm.

## Data Availability

Data are contained within this article or in the Supplementary Material.

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
