# Peer review of "Corticosterone Administration Alters White Matter Tract Structure and Reduces Gliosis in the Sub-Acute Phase of Experimental Stroke"

_ijms, 2021, doi:10.3390/ijms22136693_

Round 1

Reviewer 1 Report

The neurochemical, cellular and molecular aspects of ischemic brain injuries have not been fully understood yet and they are still  considered as a very hot topic in the contemporary neuroscience. The treatment strategies of this severe state are unfortunately  often difficult and poorly effective, so any novel possibilities are urgently required. Several intra- or intercellular events occurred in the ischemic brain regions seem to be of special interest.

This very timely, elegant and really original study by Zalewska et al. shows for the first time that corticosterone administration during rat brain ischemia may affect white matter neurohistology, axonal degeneration, and even more importantly reduced post-stroke gliosis. While results are not yet fully conclusive, the article is definitely valuable, innovative and important.

The experimental paradigm is in general very well considered photothrombic stroke technique and all  tissue processing, histology and immunohistochemistry were kept the high standard.  Appropriate statistical methods were applied with sufficient number of experimental data. The study is perfectly documented and manuscript is strongly informative. All figures are also well designed and clear. To sum up, this article may be considered as a valuable contribution to the field of experimental neuroscience. There is no doubt,  this paper may also be important and somehow intriguing for all  clinical neurologists working in the field of stroke.

I have the following suggestions for the Authors;

Figure 1,  2 and 4. Graphs should be enlarged, they are poorly visible in the present form.

Figure 5 (B). Judging by the shapes of DAPI-stained nuclei the micrographs seem to look squeezed horizontally. I suggest the picture would benefit from straingtening them.

Reviewer 2 Report

The authors present an experimental study conducted in 60 healthy male C57BL/6 mice in order to examine the changes that occur within the corpus callosum and corticospinal tract after a photothrombotic stroke, and to investigate the influence of high levels of exogenous corticosterone (CORT) administration on white matter pathology. The study  provides evidence that both CORT administration and cortical ischemia induce morphological alterations in white matter tract (WMT) structure and concomitant axonal degeneration within  the corticospinal tract. Furthermore, CORT administration after the stroke decreased stroke–induced gliosis potentially by influencing WMT reorganization. The study is interesting but some aspects of the manuscript may be improved taking into account the following points:

1.      In the Introduction the authors have certainly and appropriately mentioned that WMT hyperintensities on MRI are associated with poor performance on neuropsychological tests. Since cognitive impairment is “in vivo” an essential clinical feature of white matter leukoencephalopathy, I would suggest expanding the text in relation to the cognitive profile of white matter hyperintensities and the neuropsychological characteristics of subcortical vascular dementia  (see and comment the study published in Expert Rev Neurother 2009; 9: 1201-1217).
2.      The authors comment that their findings provide very strong indications to further investigate cortisol level and WMT structure through clinical and preclinical research. However, the authors should indicate that an indispensable line of research in the future would be precisely to perform clinical studies on WMT, cortisol level and gender, since women differ from men in the distribution of risk factors,  ischemic stroke subtype, stroke severity, and outcome (Clin Neurol Neurosurg 2014 Dec;127:19-24). The inclusion and comment of this reference is recommended.
